# Histostar-Functionalized Covalent Organic Framework for Electrochemical Detection of Exosomes

**DOI:** 10.3390/bios12090704

**Published:** 2022-09-01

**Authors:** Yuxin Lin, Beibei Nie, Xinyu Qu, Minghui Wang, Jie Yang, Genxi Li

**Affiliations:** 1State Key Laboratory of Analytical Chemistry for Life Science, School of Life Sciences, Nanjing University, Nanjing 210093, China; 2Jiangsu Co-Innovation Center of Efficient Processing and Utilization of Forest Resources, College of Science, Nanjing Forestry University, Nanjing 210037, China; 3Center for Molecular Recognition and Biosensing, School of Life Sciences, Shanghai University, Shanghai 200444, China

**Keywords:** covalent organic frameworks, Histostar, exosome, electrochemical biosensor

## Abstract

Covalent organic frameworks (COFs) are gaining growing interest owing to their various structures and versatility. Since their specific physical–chemical characteristics endow them great usage potentiality in biosensing, we herein have synthesized spherical COFs with regular shape and good dispersion, which are further used for the design of a novel nanoprobe by modifying Histostar on the surface of the COFs. Moreover, we have applied a nanoprobe for the fabrication of an electrochemical biosensor to detect exosomes. Since Histostar is a special polymer, conjugated with many secondary antibodies (IgG), and HRP can increase the availability of HRP at the antigenic site, the biosensor can have a strong signal amplification ability. Meanwhile, since COFs with high porosity can be loaded with a huge amount of Histostar, the sensitivity of the biosensor can be further improved. With such a design, the proposed biosensor can achieve a low exosomes detection limit of 318 particles/µL, and a wide linear detection range from 10^3^ particles/µL to 10^8^ particles/µL. So, this work may offer a promising platform for the ultrasensitive detection of exosomes.

## 1. Introduction

Covalent organic frameworks (COFs) are regarded as a rising kind of metal-free porous crystalline material, formed by the covalent connection and periodic extension of designed organic structural units [1,2,3,4]. Due to their diverse structure, inerratic porosity, advanced specific surface area, superior thermal stability, and adjustable pore size, COFs are considered to be promising materials with great potential in the field of energy storage [5,6,7], adsorption [8,9,10], catalysis [11,12,13], and drug delivery [14,15,16] and sensing [17,18,19,20,21]. To date, researchers extensively use solvothermal methods to synthesize COFs. Nevertheless, it is relatively difficult to control the crystal structure, uniformity, and size of the as-synthesized COFs, since the solvothermal methods typically require sealed Pyrex tubes, high temperatures, and inert atmospheres [22,23,24]. Recently, the strategies of green room-temperature synthesis for COFs have attracted great attention because they can accurately control the process of reaction, slow down the reaction rate, and easily acquire high-quality COFs [25,26].

Exosomes are vesicles with lipid bilayer membrane structures that are released by most cells and can circulate stably in body fluids [27,28,29]. Exosomes are considered to be important players in intercellular communication, and growing evidence shows that quantities of bioactive molecules are gather in exosomes and can be diverted from donor cells to recipient cells, resulting in cell-to-cell information transfer [30,31,32]. The bioactive substances in exosomes may be taken up by recipient cells, thereby promoting tumorigenesis and progression [33,34,35]. Furthermore, exosomes are not only involved in pre-metastatic niche formation, tumor angiogenesis, and tumor immunosuppression, but can also reflect the changes in the pathological and physiological states of their parental cells [36,37,38,39]. Studies have shown that the presence of the epithelial cell adhesion molecule (EpCAM) in exosomes has diagnostic value for colorectal cancer, while other markers in exosomes such as the EGFR subtypes can predict the efficacy of treatments for glioblastoma [40,41,42,43]. Furthermore, since exosomes can cross the blood–brain barrier, they may also be ideal candidates for advanced therapy in neurodegenerative diseases [44,45,46]. Due to these important functions of exosomes, it is significant to establish methods for exosome analysis with easy operation, high sensitivity, and dependability.

Herein, we have synthesized spherical COFs with a regular shape and good dispersion and have used the material to fabricate an electrochemical biosensor for exosome detection. In order to have a better performance of the biosensor, the COFs are designed to be loaded with a large amount of Histostar. Histostar is a special polymer that couples multiple secondary antibodies and HRP on one chain. This structure can increase the availability of HRP at the antigenic sites and the signal amplification ability. Meanwhile, since COFs can load a large number of Histostar due to their high porosity, and the exoskeleton of COFs can maintain the function of Histostar, the sensitivity can be further improved. So, the electrochemical biosensor fabricated in this work by using Histostar-functionalized COFs can be used for the sensitive detection of exosomes.

## 2. Materials and Methods

### 2.1. Materials and Apparatus

The 2,5-divinylterephthalaldehyde (DVA) was purchased from Jilin Chinese Academy of Sciences-Yanshen Technology Co., Ltd. (Changchun, China). Acetonitrile (ACN), tris(hydroxymethyl)aminoethane (Tris), glacial acetic acid (HAc), 1,3,5-Tris (4-aminophenyl) benzene (TPB), and tetrahydrofuran (THF) were supplied by Aladdin. The 3,3′,5,5′-tetramethylbenzidine (TMB) substrate (H_2_O_2_ included) was purchased from Sigma-Aldrich. Anti-CD63 mouse monoclonal antibody was purchased from Sangon Biotech Co., Ltd. (Shanghai, China). Anti-EpCAM rabbit monoclonal antibody was obtained from Abcam. Histostar was offered by MBL Beijing Biotech Co., Ltd. (Beijing, China). All other reagents used were analytical grade. Ultra-pure water was purified by Millipore purification system (Milli-Q, 18.2 MΩ).

The size and monodispersity of the spherical COFs were obtained using FEI Tecnai G2 F20 S-TWIN instrument and Hitachi Smur 3400N instrument. Electrochemical studies were performed by a CHI660D electrochemical workstation. Nanoparticle tracking analysis were acquired on Malvern instrument.

### 2.2. Preparation of Spherical COFs

Spherical COFs are obtained based on previous reports and with certain improvements [47]. Firstly, TPB (70 mg) and DVA (56 mg) were mingled with acetonitrile (25 mL) with 1 min ultrasonic. After that, acetic acid (5 mL, 12 M) was continuously dripped into the solution and oscillate violently for 10 s. Then, the solution was placed at room temperature for 3 days. The solids were collected via 9500 rpm centrifugation, washed several times, and dried in vacuum at 60 °C.

### 2.3. Preparation of Histostar@COFs

Histostar@COFs was prepared by combining 1 mL of Histostar with COFs dispersion (1 mL, 0.1 mg mL^−^^1^) and stirring for 4 h. Then, the product was obtained via centrifugation (9500 rpm) and finally dispersed in PBS (1 mL, 10 mM).

### 2.4. Cell Culture and Exosome Isolation

The cells were cultured in 1640 medium containing 1% (*v*/*v*) penicillin-streptomycin and 10% (*v*/*v*) FBS under a moist environment with 5% CO_2_. When the cells grew to 70%, the supernatant was discarded and the cells were transferred into serum-free medium for two days. Finally, the treated supernatant was collected for exosome isolation.

Centrifuge (2000× *g*, 20 min) the collected supernatant to remove cell fragments and larger molecules. Then, centrifuge at 11,000× *g* for 0.5 h, and then ultrafiltration (0.22 μm pore diameter). The exosomes were obtained by centrifuging the solution at 120,000× *g* for 2 h, and the obtained exosome was kept in −80 °C for further use. Nanoparticle tracking analysis (NTA) was used to measure the concentration of exosome.

### 2.5. Preparation of the Biosensor

The gold electrodes were pretreated based on the previous literature [48]. The anti-CD63 mouse monoclonal antibody (Ab1) was immobilized onto the gold electrodes via thiol-modification of primary amines (-NH_2_) of the Ab1 to introduce sulfhydryl (-SH) groups, thus allowing covalent immobilization of the thiolated antibodies to the gold electrode (the electrode was labeled as Ab1/Au). After washing with PBS, the electrode was treated with 3% BSA for 1 h in order to prevent non-specific adsorption. Next, 5 μL of exosomes at different concentrations were dropped onto the Ab1/Au and incubated at 37 °C for 2 h. After washing with PBS, 5 μL of 0.5 mg mL^−^^1^ anti-EpCAM rabbit monoclonal antibody (Ab2) was dripped onto the exosome/Ab1/Au electrode surface and incubated at 37 °C for 1 h. The prepared electrode was labeled as Ab2/exosome/Ab1/Au. After washing PBS for three times, the prepared Histostar@COFs was dripped onto the Ab2/exosome/Ab1/Au and incubated for 1 h, and the electrode was labeled as Histostar@COFs/Ab2/exosome/Ab1/Au.

### 2.6. Electrochemical Detection

Electrochemical impedance spectroscopy (EIS) and amperometric i-t curves were measured by a CHI660D electrochemical workstation. Gold electrodes, saturated calomel electrodes (SCE), and platinum electrodes were used as working electrodes, reference electrodes, and counter electrodes, respectively. EIS was measured in 0.1 M PBS buffer containing 5.0 mM [Fe(CN)_6_]^3−/4−^ and 0.1 M KCl (pH = 7.4) with a ranging frequency from 0.1 to 10^5^ Hz. The i-t curves were measured in a TMB substrate solution (H_2_O_2_ included) at a voltage of −0.1 V.

## 3. Results and Discussion

### 3.1. Principle of the Biosensor

Figure 1 depicts the preparation process of the COFs-based nanoprobes as well as the principle of the proposed biosensor for exosome detection. Firstly, COFs are synthesized by TPB and DVA at room temperature. Then, a large amount of Histostar is loaded on the surface of spherical COFs. Histostar is a special polymer that couples multiple secondary antibodies and HRP on one chain. This structure can increase the availability of HRP at antigen sites and, thus, has a very effective signal amplification effect. Meanwhile, due to their high porosity, COFs are able to carry a huge amount of Histostar, and the stability of Histostar can be significantly improved by the exoskeleton of COFs. For the detection of the exosome, the anti-CD63 mouse monoclonal antibody is firstly immobilized on the surface of the electrode. After exosomes are captured on the electrode, the anti-EpCAM rabbit monoclonal antibody will combine with the EpCAM on the exosome surface. After that, through the specific recognition between primary and secondary antibody, the captured anti-EpCAM rabbit monoclonal antibody can further combine with the secondary antibody IgG in Histostar@COFs, thus introducing a large amount of HRP to the electrode surface. Under the action of TMB and H_2_O_2_, strong electrochemical signals can be produced for the quantitative detection of exosomes.

### 3.2. Characterization of Materials

We have characterized the size and morphology of the prepared nanomaterials through SEM and TEM. Figure 2A,B exhibits the typical SEM image (Figure 2A) and the TEM image (Figure 2B) of the spherical COFs. The prepared COFs are of uniform sphere and have good dispersibility. The average diameter is about 450 nm. To verify the performance of the material, the comparison of Histostar@COFs and HRP-labeled mouse anti-rabbit IgG (HRP-IgG) has been provided, and the results are shown in Figure 2C. The experimental results indicate that the current response in the presence of Histostar@COFs is significantly higher than that of HRP-IgG, which suggests that Histostar@COFs can effectively amplify the signal.

### 3.3. Feasibility Verification of the Method

The preparation process of this biosensor has been characterized by EIS. As shown in Figure 3A, the bare gold electrode (curve a) possesses a negligible semicircle, which indicates that the charge transfer resistance (R_ct_) is very low. When Ab1 (curve b), exosomes (curve c), and Ab2 (curve d) are modified on the electrode surface, the R_ct_ values increase significantly, because the electron transfer of [Fe(CN)_6_]^3−/4−^ is hindered by the antibodies and exosomes. When Histostar@COFs (curve e) are modified on the electrode, the R_ct_ further increases, indicating the successful combination between the exosomes and Histostar@COFs. In addition, we have measured the current response of different modified electrodes to confirm the preparation of this biosensor. As shown in Figure 3B, since the Histostar@COFs nanoprobe cannot bind to the electrode, the bare electrode (curve a) and electrode without exosomes (curve b) only exhibit weak currents. Nevertheless, when exosomes are present in the sample, the current increases significantly (curve c). The reason is that a large amount of HRP has been introduced to the electrode surface, which can generate a strong electrochemical signal. So, this method is feasible for the detection of exosomes.

### 3.4. Analytical Performance of the Biosensor

The proposed biosensor has been used for the analysis of exosomes at various concentrations. Experimental results reveal that the current increases with the increase in exosome concentration from 10^3^ to 10^8^ particles/µL (Figure 4A). There is a linear relationship between the current response and the logarithm of the exosome concentration (Figure 4B). The regression equation is I = 0.4140 logc—0.7615, and the correlation coefficient is 0.9952. The limit of detection is 318 particles/µL (S/N = 3). Compared with some other reports for exosome detection (Table 1), this method possesses a wider linear range and a relatively low detection limit, suggesting its excellent performance.

To assess the selectivity of this sensor, we have selected some substances that may coexist in biological samples as interferences, such as cysteine, glucose, glutathione, and BSA (the concentration of exosomes is 10^6^ particles/μL, and the concentrations of interferents are 1 mM). As shown in Figure 5A, compared with the other three interferers, the current response for exosomes is significantly higher, indicating that the biosensor has good selectivity for exosome detection in biological samples. In addition, we have evaluated the reproducibility of this biosensor through relative standard deviation (RSD). Six tests have been conducted, respectively, with the concentration of exosomes as 10^6^ particles/μL (Figure 5B). The RSD of the six tests is 2.27%, indicating the good reproducibility of this biosensor.

## 4. Conclusions

In this work, we have synthesized spherical COFs with a regular shape at room temperature by the Schiff base reaction between DVA and TPB. Through the reasonable control of the reaction conditions, the obtained COFs present good crystallinity and morphology, a large specific surface area, and good stability. After preparation, the COFs have been further functionalized with Histostar and then applied for the sensitive detection of exosomes. So, this work may open up a new avenue for the application of COFs in biosensing, while it also provides a simple and effective method for exosome detection. The simple and easy preparation method of spherical COFs as well as their functionalization by Histostar may advance the development of biosensors based on metal-free porous crystalline materials.

## Figures and Tables

**Figure 1 biosensors-12-00704-f001:**
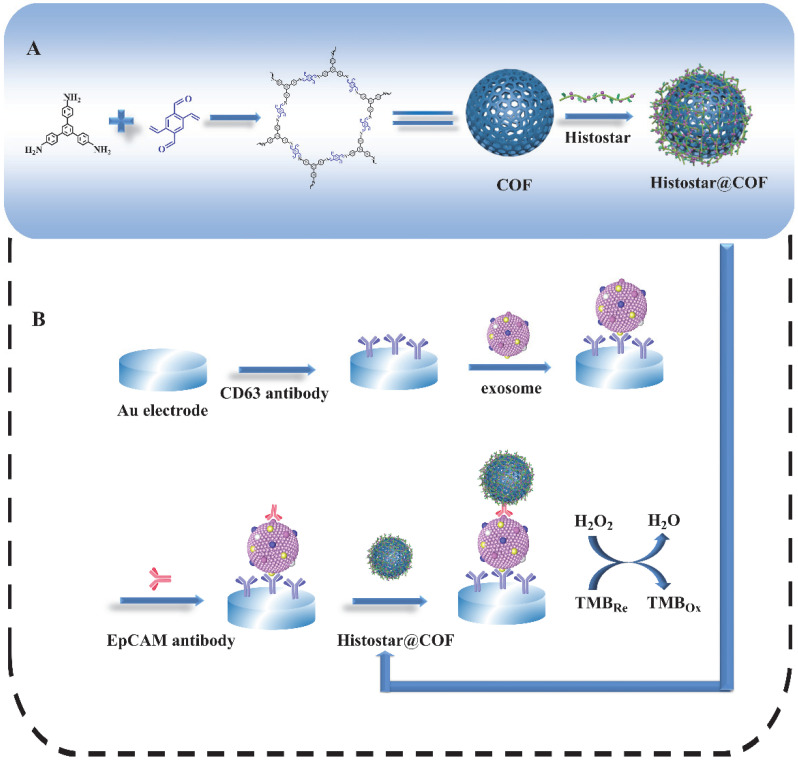
Schematic graph for (**A**) the preparation process of Histostar@COFs and (**B**) the principle of the proposed biosensor for the detection of exosomes.

**Figure 2 biosensors-12-00704-f002:**
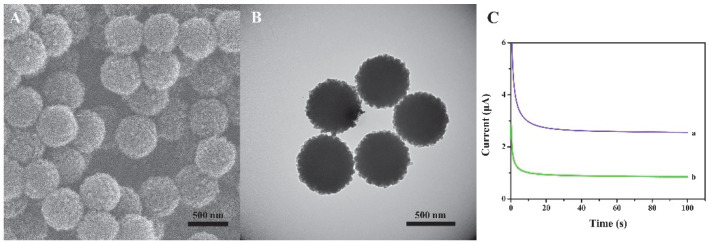
SEM(**A**) and TEM (**B**) images of COFs. (**C**) The current response of (a) Histostar@ COFs and (b) HRP-IgG.

**Figure 3 biosensors-12-00704-f003:**
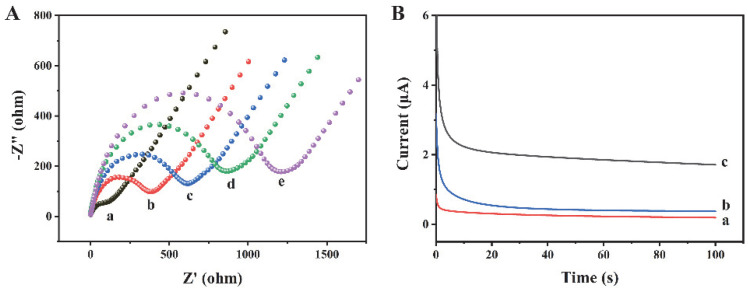
(**A**) EIS measurements of (a) bare gold electrode, (b) Ab1/Au, (c) exosomes/Ab1/Au, (d) Ab2/exosome/Ab1/Au, and (e) Histostar@COFs/Ab2/exosome/Ab1/Au. (**B**) Currents of different electrodes in detection buffer: (a) bare Au, (b) Histostar@COFs/Ab2/exosome/Ab1/Au when the exosome concentration was 0 particles/μL, and (c) Histostar@COFs/Ab2/exosome/Ab1/Au with 10^6^ particles/μL concentration exosomes.

**Figure 4 biosensors-12-00704-f004:**
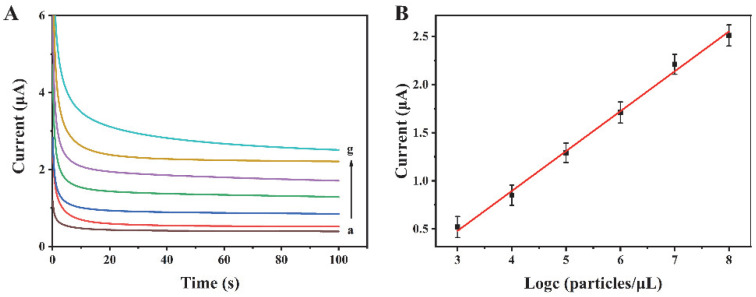
(**A**) Current curves for different concentrations of exosomes. a–g are 0, 10^3^, 10^4^, 10^5^, 10^6^, 10^7^, and 10^8^ particles/μL, respectively. (**B**) The linear relationship between the current intensity and the logarithm of exosome concentration.

**Figure 5 biosensors-12-00704-f005:**
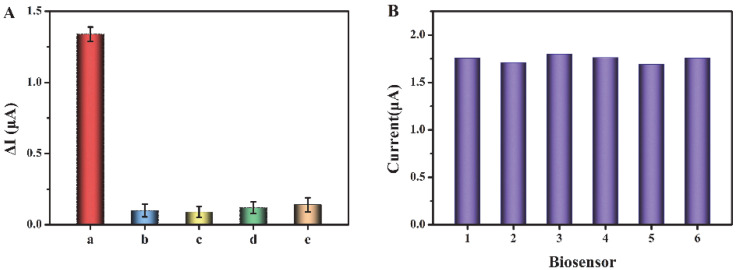
(**A**) Current change (ΔI) before and after addition of exosomes and interferents. a–e are exosome, cysteine, glucose, glutathione, and BSA, respectively. (**B**) Reproducibility of the biosensor.

**Table 1 biosensors-12-00704-t001:** Comparison of the developed method with some other reports.

Technique	Linear Range Particles/µL	Detection Limit Particles/µL	Reference
Fluorescence	8.5 × 10^3^ to 8.5 × 10^5^	4.5 × 10^3^	[49]
Fluorescence	10^5^ to 10^9^	10^5^	[50]
Fluorescence	1.1 × 10^4^ to 1.1 × 10^7^	562	[51]
Surface plasmon resonance	0.1 to 10^7^	10^4^	[52]
Electrochemiluminescence	10^3^ to 10^6^	400	[53]
Colorimetry	2.0 × 10^3^ to 5.0 × 10^5^	1.2 × 10^3^	[54]
Colorimetry	10^8^ to 10^10^	1.5 × 10^8^	[55]
Electrochemistry	10^4^ to 10^7^	9661	[56]
Electrochemistry	10^3^ to 10^8^	318	This work

## Data Availability

Not applicable.

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
