# Peer review of "Histostar-Functionalized Covalent Organic Framework for Electrochemical Detection of Exosomes"

_biosensors, 2022, doi:10.3390/bios12090704_

Round 1
Reviewer 1 Report
The manuscript deals with a detailed characterization study performed on spherical Covalent organic frameworks (COFs) synthesized with regular shape and good dispersion, whose specific physical-chemical characteristics make them greatly used in biosensing. The authors uses the obtained material to fabricate an electrochemical biosensor for exosome detection, increasing the performance of the biosensor by loading the COFs with a large amount of Histostar, a special polymer that couples multiple secondary antibodies and HRP on one chain. The size and morphology of the prepared nanomaterials were characterized through SEM and TEM techniques.
The manuscript is clearly written in each part resulting easy to read even for non-expert readers. The Figures are clear, I have particularly appreciated Figure 1 that displays a schematic graph for the preparation process of COFs functionalized with Histostar and the “principle” of the proposed biosensor for the detection of exosomes.
I have actually only this observation: may the authors describe a "real" application of the proposed nanomaterials as biosensor for the detection of exosomes. Do they have “experimental evidence” that this material work? If the answer is yes, then I suggest adding this information since this would give more strength to all the work; otherwise, they could in any case justify why they do not have yet this information.
In conclusion, I think that the manuscript deserves the publication on the journal Biosensor after this major revisions.
Author Response
Many thanks for the helpful comment and suggestion. The “experimental evidence” that this material work has been presented in Figure 2C, and the relevant description of this result has also been added in the revised manuscript (lines 151-158).
Reviewer 2 Report
Covalent organic frameworks (COFs) are relatively new and emerging materials with multiple applications in various fields. In the recent years COFs are widely used for preparation of fluorescent or electrochemical sensors. In present communication authors describe simple preparation of spherical COF nanoparticles coated with Histostar polymer and the usage of such hybrid for electrochemical detection of exosomes with antibody-modified electrode. Authors showed, that this method possessed good sensitivity and linearity in broad range of exosomes concentration. Also high selectivity to exosomes was demonstrated in comparison with small molecules. However, I have several notes on this manuscript:
1) References in introduction should be associated with modern relevant and representative reviews in the area of COFs and COF-based sensors. Authors gave a few references to reviews (Refs [11], [12], [19]), but some of modern reviews were not cited:
https://www.mdpi.com/1424-8220/22/13/4758
https://pubs.acs.org/doi/10.1021/acs.chemrev.9b00550
https://www.sciencedirect.com/science/article/pii/S0010854521002319
https://www.frontiersin.org/articles/10.3389/fchem.2020.601044/full
2) Authors checked selectivity of the sensor toward exosomes against small molecules. To my opinion, authors also should check selectivity in comparision with biopolimers, such as proteins or nucleic acids.
3) Figures 1,3,4,5 have too low resolutions. Authors should enhance quality of the graphics.
Overall, the manuscript could be published after this minor revisions.
Author Response
Many thanks for the helpful comment and suggestion. We have carefully revised this manuscript according to your help. Here is the point-by-point response to your comment and suggestion.
Covalent organic frameworks (COFs) are relatively new and emerging materials with multiple applications in various fields. In the recent years COFs are widely used for preparation of fluorescent or electrochemical sensors. In present communication authors describe simple preparation of spherical COF nanoparticles coated with Histostar polymer and the usage of such hybrid for electrochemical detection of exosomes with antibody-modified electrode. Authors showed, that this method possessed good sensitivity and linearity in broad range of exosomes concentration. Also high selectivity to exosomes was demonstrated in comparison with small molecules. However, I have several notes on this manuscript:
1) References in introduction should be associated with modern relevant and representative reviews in the area of COFs and COF-based sensors. Authors gave a few references to reviews (Refs [11], [12], [19]), but some of modern reviews were not cited:
https://www.mdpi.com/1424-8220/22/13/4758
https://pubs.acs.org/doi/10.1021/acs.chemrev.9b00550
https://www.sciencedirect.com/science/article/pii/S0010854521002319
https://www.frontiersin.org/articles/10.3389/fchem.2020.601044/full
Many thanks for the helpful suggestion. The relevant modern reviews have been cited in this revised manuscript (refs. 3, 18-20). Accordingly, the sequence number of the other references are changed.
2) Authors checked selectivity of the sensor toward exosomes against small molecules. To my opinion, authors also should check selectivity in comparision with biopolimers, such as proteins or nucleic acids.
Many thanks for the helpful comment and suggestion. We have checked selectivity in comparision with proteins in the revised manuscript (Figure 5A)
3) Figures 1,3,4,5 have too low resolutions. Authors should enhance quality of the graphics.
Many thanks for the helpful suggestion. We have enhanced quality of the graphics.
Overall, the manuscript could be published after this minor revisions.
Many thanks for the positive comments and kind recommendation.
Reviewer 3 Report
Comments to Authors
I have evaluated the manuscript “Histostar-functionalized covalent organic framework for electrochemical detection of exosomes”. The manuscript discusses the modification of COF with histostar for electrochemical detection of exosomes. I feel that a major revision is necessary to proceed further. Please find my comments.
-
Please include the experimental conditions for the Ab1 incubation on gold electrode
-
Please make the Fig. 1 clear. The captions are not visible
-
The authors states that “COFs are able to carry a huge number of Histostars”. Is there any quantitative measure has been done for loading of histostars on COFs?
-
There is no continuity in Fig 1B with upper portion and lower portion. Please include an arrow to show the continuity.
-
Figure caption 3 B is little bit confusing. It is given that Histo star@COFs/Ab2/exosome/Ab1/Au without exosomes, . Its like exosome without exosomes. Please justify.
-
What is the concentration of exoseme and interferents used for the selectivity study ? Please include the details
-
The selectivity data looks good when the authors plot the current change (with respect to background) instead of current.
-
A comparison of sensor performance with the other existing sensor is highly recommended.
-
In the sentence “Nevertheless, when exosomes are present in the sample, the current increases significantly (curve c), verifying that this method for exosome detection is feasible. What is the mechanism behind the current increase. Please include this in one or two sentences.
-
Electron transfer resistance (Ret) is same as charge transfer resistance (Rct)?. Normally Rct is preferred.
-
Regeneration of the sensor surface is possible here? Or it is single use?
-
The authors have concluded that “the obtained COFs present good crystallinity and morphology, large specific surface area, and good stability ”. Is there any data to prove this?
Author Response
Many thanks for the helpful comment and suggestion. We have carefully revised this manuscript according to your help. Here is the point-by-point response to your comment and suggestion.
I have evaluated the manuscript “Histostar-functionalized covalent organic framework for electrochemical detection of exosomes”. The manuscript discusses the modification of COF with histostar for electrochemical detection of exosomes. I feel that a major revision is necessary to proceed further. Please find my comments.
1. Please include the experimental conditions for the Ab1 incubation on gold electrode
Many thanks for the helpful suggestion. The experimental conditions for the Ab1 incubation on gold electrode have been included in the revised manuscript (lines 104-108).
2. Please make the Fig. 1 clear. The captions are not visible
Many thanks again for the helpful suggestion. So, we have made the Fig. 1 clear.
3. The authors states that “COFs are able to carry a huge number of Histostars”. Is there any quantitative measure has been done for loading of histostars on COFs?
Many thanks for the helpful comment. The quantitative measure has not been done for loading of histostars on COFs.
4. There is no continuity in Fig 1B with upper portion and lower portion. Please include an arrow to show the continuity.
Many thanks for the helpful comment and suggestion. So, we have included an arrow to show the continuity.
5. Figure caption 3 B is little bit confusing. It is given that Histo star@COFs/Ab2/exosome/Ab1/Au without exosomes, . Its like exosome without exosomes. Please justify.
Many thanks for the help. We have corrected the description of Figure caption 3 B (lines 179).
6. What is the concentration of exoseme and interferents used for the selectivity study ? Please include the details
Many thanks for the helpful suggestion. The concentration of exoseme and interferents used for the selectivity study have been include in the revised manuscript (lines 197-198).
7. The selectivity data looks good when the authors plot the current change (with respect to background) instead of current.
Many thanks for the help. We have plotted the current change (with respect to background) instead of current in the revised manuscript (Figure 5A).
8. A comparison of sensor performance with the other existing sensor is highly recommended.
Many thanks for the helpful suggestion. So, a comparison of sensor performance with the other existing sensor has been added in the revised manuscript (Table 1). The corresponding information has also been added in this revised manuscript (lines 191-193).
9. In the sentence “Nevertheless, when exosomes are present in the sample, the current increases significantly (curve c), verifying that this method for exosome detection is feasible. What is the mechanism behind the current increase. Please include this in one or two sentences.
Many thanks for the helpful suggestion. The mechanism behind the current increase has been added in the revised manuscript (lines 172-174)
10. Electron transfer resistance (Ret) is same as charge transfer resistance (Rct)?. Normally Rct is preferred.
Many thanks for the help. So, Rct, instead of Ret, has been used in the revised manuscript (lines 162, 163, 166).
11. Regeneration of the sensor surface is possible here? Or it is single use?
Many thanks for the helpful comment. The sensor surface is single use here.
12. The authors have concluded that “the obtained COFs present good crystallinity and morphology, large specific surface area, and good stability ”. Is there any data to prove this?
Many thanks again for the helpful comment. We have characterized the obtained COFs through transmission electron microscopy and scanning electron microscopy (Figure 2).
Round 2
Reviewer 1 Report
The authors have followed all the suggestions hence the manuscript deserves publication on the jounal Biosensors in the present form.
Reviewer 3 Report
Based on the author's revision, the manuscript can be accepted in the present form.